# Problematic Internet Use in Adolescents Is Driven by Internal Distress Rather Than Family or Socioeconomic Contexts: Evidence from South Tyrol, Italy

**DOI:** 10.3390/bs15111534

**Published:** 2025-11-11

**Authors:** Christian J. Wiedermann, Verena Barbieri, Giuliano Piccoliori, Adolf Engl

**Affiliations:** Institute of General Practice and Public Health, Claudiana College of Health Professions, 39100 Bolzano, Italy

**Keywords:** problematic Internet use, adolescents, psychological distress, digital behavior, mental health, emotional regulation, South Tyrol, cross-sectional survey

## Abstract

Problematic Internet use is an emerging concern in adolescent mental health and is closely linked to psychological distress and emotional regulation. This cross-sectional study analyzed self-reported data from 1550 adolescents aged 11–19 years in South Tyrol, a linguistically and culturally diverse region in Northern Italy. Problematic Internet use was measured using the validated Generalized Problematic Internet Use Scale 2 (GPIUS-2), along with standardized instruments for depressive symptoms (PHQ-2) and anxiety (SCARED-GAD). Multivariable regression analysis revealed that depression and anxiety symptoms were the strongest independent predictors of higher GPIUS-2 scores. In contrast, demographic factors such as gender, family language, urbanization, migration background, and parental education were not significantly associated with PIU. Modest associations were observed between GPIUS-2 scores and both perceived economic burden and parental use of digital control tools. Perceived family support showed a small protective effect. These findings underscore the central role of emotional vulnerability in adolescent PIU and suggest that interventions should focus on supporting mental health and adaptive coping rather than solely targeting screen time or structural family characteristics.

## 1. Introduction

The pervasive integration of digital technologies into everyday life has reshaped adolescents’ socialization, education, and leisure behaviors worldwide. Alongside the manifold opportunities provided by the Internet, concerns have grown about problematic Internet use (PIU) among youth, a behavioral pattern characterized by excessive or poorly controlled preoccupations, urges, or behaviors regarding Internet use that lead to impairment or distress ([32]; [30]). PIU in adolescence is associated with a spectrum of psychosocial detriments, including academic difficulties, social withdrawal, and mental health problems such as depression, anxiety, and suicidality ([35]; [30]; [29]).

Theoretical frameworks help us understand why some adolescents develop PIU while others do not. The Compensatory Internet Use Theory (CIUT) ([31]) is a conceptual model suggesting that individuals may engage in PIU to compensate for offline psychosocial difficulties, including psychological distress, family dysfunction, and socioeconomic hardships. The Internet provides a means of escaping negative emotions and real-world stressors. This aligns with findings linking PIU to depression, anxiety, poor self-esteem, and family adversity ([35]; [30]; [29]).

While global research has delineated several consistent risk factors for adolescent PIU, including male gender, older age, lower socioeconomic status, poor family functioning, and psychological vulnerabilities ([23]; [43]; [49]), regional studies remain scarce, especially in culturally unique territories such as South Tyrol in northern Italy. South Tyrol is a bilingual and multicultural region of Northern Italy, with German-, Italian-, and Ladin-speaking populations, as well as urban and rural differences ([46]). While these factors are often assumed to shape adolescents’ psychosocial experiences, empirical evidence from this region remains limited. Evidence from surveys during the COVID-19 pandemic underscored the vulnerability of adolescents to psychosocial stressors, highlighting the association between psychological distress and various health-compromising behaviors ([8], [7]). However, these prior waves did not examine problematic digital behaviors in depth, despite the evident surge in online activity during and after the pandemic.

The “Corona and Psyche South Tyrol” (COP-S) 2025 survey offers a perspective for exploring how adolescents in South Tyrol’s cultural context may use digital environments in response to perceived burdens. The fourth wave of this longitudinal research program newly integrated validated measures of problematic Internet use (GPIUS-2) and psychological distress (PHQ-2, SCARED-GAD), alongside indicators of perceived economic strain, family support, and parental digital mediation. Conducted bilingually (German and Italian) across all school types, the COP-S design enables culturally sensitive analysis of adolescent digital behavior and its psychosocial determinants within the South Tyrolean setting. By adopting CIUT ([31]), this study examines PIU as a coping strategy shaped by contextual stressors, investigating how psychological distress, family dynamics, and socioeconomic factors influence PIU among South Tyrol adolescents (Figure 1).

Building on international research and the regional need for population-based insights, this study investigated the determinants of PIU in South Tyrolean adolescents by focusing on three domains consistently identified in the literature:PIU is closely linked to internalizing symptoms, such as depression and anxiety ([35]; [30]; [29]). However, it remains unclear how these associations manifest in the specific psychosocial climate of post-pandemic South Tyrol.Poor parent-child relationships, low family support, and limited parental involvement have been repeatedly associated with an elevated PIU risk ([38]; [49]). The protective role of perceived family support has not yet been examined in this setting.While international studies have linked low SES and parental unemployment to problematic digital behavior ([23]; [43]; [49]), previous COP-S surveys in South Tyrol have not systematically assessed associations between socioeconomic status indicators and youth psychosocial outcomes. The 2025 wave of COP-S includes, for the first time, adolescent-reported subjective economic burden, allowing for a more nuanced analysis of perceived financial stress as a potential risk factor for depression.

This study investigates the following research questions: (i) How are depression and anxiety associated with PIU among South Tyrolean adolescents? (ii) To what extent do perceived family support and parent-child relationship quality predict PIU, and how might these associations vary across language groups or between urban and rural settings? (iii) What are the roles of subjective economic burden and parental education in shaping adolescents’ risk of PIU?

Based on the Compensatory Internet Use Theory (CIUT) and prior international findings, the following hypotheses were formulated.

Higher levels of depressive and anxiety symptoms are positively associated with higher problematic Internet use (GPIUS-2 total score).Greater perceived family support is negatively associated with problematic Internet use.Subjective financial burden is positively associated with problematic Internet use, whereas structural sociodemographic factors (age, gender, parental education, family affluence, migration background, family language, and urbanity) show weak or no associations.

By addressing these hypotheses, this study aims to generate evidence-based insights to inform digital health promotion and targeted prevention strategies for adolescents in South Tyrol.

## 2. Methods

### 2.1. Study Design and Sample

This cross-sectional analysis was based on data from the fourth wave of the COP-S survey series, conducted between 17 March and 13 April 2025. The anonymous online survey targeted families with school-aged children in South Tyrol. Recruitment was facilitated through public school directorates, which distributed personalized survey links via email to approximately 40,000 families.

The survey was administered using the SoSci Survey platform (Version 3.2.46; SoSci Survey GmbH, Munich). It retained the core structure of previous COP-S waves ([8], [7]), while incorporating new instruments to assess post-pandemic psychosocial stressors and digital behavior patterns among adolescents.

For the present analysis, adolescents aged 11–19 years who completed the self-report section of the questionnaire were included. One parent per adolescent independently completed the corresponding proxy report version. Parental consent and adolescent assent were obtained prior to their participation. The sample approximates the regional distribution of school-aged youth by age and gender based on official provincial statistics. No quota sampling or weighting procedures were applied. Representativeness by age and gender was descriptively verified against official provincial data from the Provincial Institute for Statistics (ASTAT), showing close correspondence between the sample and the underlying population distribution. Family language, as reported by parents, was used in subgroup analyses to reflect South Tyrol’s multilingual context; no statistical weighting was applied to this variable.

### 2.2. Measures

To reduce respondent burden while maintaining international comparability, the survey employed brief, psychometrically validated screening instruments widely used in adolescent research (e.g., PHQ-2, SCARED-GAD, MSPSS-Family, GPIUS-2). These short forms balance feasibility in school settings with established reliability and construct validity.

#### 2.2.1. Sociodemographic and Cultural Variables

Adolescents self-reported their age and sex. The language group was derived from parental reports of the primary language spoken at home with the child. Parental education was reported in the parent questionnaire and classified into three categories using the Comparative Analysis of Social Mobility in Industrial Nations (CASMIN) framework: low (primary and lower secondary education), medium (upper secondary education), and high (tertiary education) ([14]). Urbanity was categorized based on official zoning classifications provided by the provincial administration of South Tyrol.

Family affluence was assessed using the Family Affluence Scale III (FAS III) ([13]; [20], [19]). The 6-item scale included questions on household car ownership, whether the child had a bedroom of their own, the number of computers or tablets, the number of bathrooms, the presence of a dishwasher, and the frequency of family holidays in the previous year. Each item was coded according to the international scoring guidelines, resulting in a total sum score ranging from 0 to 13, with higher values indicating greater affluence. In the present study, the FAS III was analyzed both as a continuous variable and categorized into low, medium, and high affluence based on sample-specific tertile cut-offs ([53]).

The subjective socioeconomic burden was assessed using a single-item measure of perceived financial strain due to rising prices (“perceived inflation”), rated on a 5-point Likert scale ranging from 1 (“not at all burdensome”) to 5 (“extremely burdensome”). Information on family structure (e.g., single-parent households) and migration background was obtained from the parents’ responses.

#### 2.2.2. Psychological Distress

Adolescents reported psychological distress using two screening instruments.

Depressive symptoms were measured using the Patient Health Questionnaire-2 (PHQ-2), a 2-item screener validated in adolescent populations: “Over the last two weeks, how often have you been bothered by the following problems: (1) little interest or pleasure in doing things, and (2) feeling down, depressed, or hopeless?” Items are rated on a 4-point Likert scale ranging from 0 (“not at all”) to 3 (“nearly every day”); a total score of ≥3 indicates elevated depressive symptoms. Cronbach’s α is typically 0.79–0.83 in adolescent and adult samples ([33]; [21]; [50]).Anxiety symptoms were assessed using the Generalized Anxiety Disorders (GAD) subscale of the Screen for Child Anxiety-Related Emotional Disorders (SCARED), consisting of nine items rated on a 3-point scale from 0 (“not true”) to 2 (“very or often true”). A score of ≥9 indicates clinically relevant anxiety symptoms. Cronbach’s α is around 0.84–0.88; retest reliability r ≈ 0.70–0.80 ([10]; [18]; [55]).

#### 2.2.3. Family Dynamics

Perceived family support and quality of the parent-child relationship were assessed using the Multidimensional Scale of Perceived Social Support (MSPSS). The MSPSS includes 12 items rated on a 7-point Likert scale (1 = “strongly disagree” to 7 = “strongly agree”) and covers three subscales: family, friends, and significant others. Cronbach’s α is 0.87–0.90 for the family subscale; total scale α is around 0.88–0.92 across cultural adaptations ([57]; [1]; [11]). For this study, the family subscale was used as an indicator of emotional support and the perceived reliability of family members as a source of help. Higher scores reflect stronger perceived support from the family.

Perceived fairness in the parent-child relationship was assessed using a self-developed item (“My parents treat me fairly”), rated on a 5-point Likert scale from 1 (“never”) to 5 (“always”).

Parental involvement in school-related matters was assessed again using a self-developed item (“How often have your parents helped you during the current school year when you had a problem with school-related issues?”). Responses were recorded on a 5-point Likert scale ranging from 1 (“never”) to 5 (“always”).

The use of family based parental control tools was assessed via a parent-reported item asking whether digital supervision or organizational apps were used when the child accessed digital media. Specifically, parents were asked, “Do you use parental control settings (e.g., Family Link, Apple Family Sharing, TimeLimit, etc.) when your child uses digital media?”. The response options were binary (yes/no). Responses were analyzed as binary indicators of family level digital engagement.

#### 2.2.4. Problematic Internet Use (Primary Outcome)

PIU was assessed using the Generalized Problematic Internet Use Scale 2 (GPIUS-2), a validated 15-item self-report instrument designed to measure compulsive and maladaptive patterns of Internet use ([15]). The scale comprises five sub-dimensions:Preference for online social interaction (e.g., “I prefer online social interaction to face-to-face communication”).Mood regulation (e.g., “I use the Internet to feel better when I am down”).Cognitive preoccupation (e.g., “I think obsessively about going online when I am offline”).Compulsive Internet use (e.g., “I have difficulty controlling the amount of time I spend online”).Negative outcomes (e.g., “My Internet use has created problems for me in my life”).

Higher total scores indicate greater problematic Internet use. In the original validation study, the GPIUS-2 demonstrated excellent internal consistency (Cronbach’s α = 0.91) and strong factorial validity ([15]). The German version showed high internal consistency (α = 0.91 in an online sample; α = 0.86 in an offline sample) and good test–retest reliability (r_tt = 0.85) ([9]). The Italian version, validated in adolescents and young adults, showed Cronbach’s alpha values between 0.78 and 0.89 across subscales, and confirmatory factor analysis supported the original structure ([25]).

### 2.3. Statistical Analysis

Descriptive statistics were calculated for all study variables. Continuous data were summarized using means and standard deviations, and categorical data were described using frequencies and percentages.

Bivariate analyses explored the associations between PIU and each predictor domain (psychological distress, family support, and socioeconomic status indicators) using Spearman’s rho for continuous variables and chi-square tests for categorical comparisons. The relationship between parental use of digital control tools (yes/no) and GPIUS-2 scores was examined using independent-sample *t*-tests to compare mean values between groups.

To identify independent predictors of problematic Internet use, a multiple linear regression model was constructed using the GPIUS-2 total score as the dependent variable. Associations between the GPIUS-2 scores and sociodemographic or contextual variables (e.g., migration background, family structure, residential environment, parental education, financial burden, and digital family communication) were first examined using bivariate analyses (*t*-tests, one-way ANOVA, or Pearson’s r). Variables showing significant or near-significant associations (*p* < 0.10) were then included in the multivariable regression model:Depressive symptoms (PHQ-2 score)Anxiety symptoms (SCARED-GAD score)Perceived family support (MSPSS family subscale)Subjective economic burdenUse of digital parental controls (Yes/No)Age group (11–14 vs. 15–19 years)Gender (male vs. female)

Multicollinearity was assessed using variance inflation factors (VIF < 5). The model fit was evaluated using the adjusted R^2^ and residual diagnostics. Statistical significance was set at *p* < 0.05 (two-tailed).

All analyses were conducted using IBM SPSS Statistics (version 27.0; IBM Corp., Armonk, NY, USA). Statistical significance was set at *p* < 0.05, with Bonferroni corrections applied for multiple comparisons.

## 3. Results

### 3.1. Sample Characteristics

A total of 2554 cases of consented self-reports were included. The total GPIUS-2 score was available for 1550 patients, ranging from 15 to 120. The sample characteristics are presented in Table 1.

The adolescent sample (*n* = 1550) was balanced in terms of age and gender, with similar proportions of younger (11–14 years) and older (15–19 years) participants, and equal representation of males and females. The vast majority came from German-speaking families, with smaller groups from Italian- and Ladin-speaking households. Most adolescents lived in two-parent households and had no migration background.

Parental education was evenly distributed across low, medium, and high educational levels. While most participants resided in rural areas, urban dwellers were also well represented.

A large share of adolescents reported feeling strongly to extremely burdened by rising prices, reflecting subjective economic stress, even though objective material wealth as measured by the FAS III was mostly moderate to high.

Among 11- to 17-year-olds, about half reported that their parents used digital control tools such as screen-time apps. Most adolescents received at least occasional help from their parents with school-related matters.

### 3.2. Age and Gender Differences in Problematic Internet Use

To explore developmental patterns, Figure 2 illustrates the age-related distribution of the mean GPIUS-2 total and subscale scores across the adolescent sample. The top panel shows a steady increase in the total GPIUS-2 score across ages 11 and 19, with the highest levels observed in the oldest age group. The bottom panel breaks down this trend across the five sub-dimensions. All subscale scores, including Preference for Online Social Interaction, Mood Regulation, Cognitive Preoccupation, Compulsive Internet Use, and Negative Outcomes, showed a gradual upward trend with increasing age, suggesting a consistent intensification of problematic Internet use behaviors during adolescence. Mood Regulation and Preference for Online Social Interaction showed the steepest increases over time.

Descriptive statistics for the GPIUS-2 total score and its five subscales are presented in Table 2, stratified by sex and age group. The mean total GPIUS-2 score across the full adolescent sample was 39.61 (SD 19.49). When comparing age groups, adolescents aged 15–19 years consistently scored higher across all GPIUS-2 domains than those aged 11–14 years, with statistically significant differences observed for the total score and all subscales (*p* < 0.001 or *p* = 0.001, respectively).

No significant sex differences were found in the total GPIUS-2 score or in most subscales, except for the Mood Regulation subscale, where females scored significantly higher than males (10.18 vs. 8.97; *p* < 0.001), indicating a stronger tendency among girls to use the Internet for emotional regulation.

These findings suggest that older adolescents and female adolescents (for mood regulation) may be at a higher risk for problematic Internet use patterns, particularly in domains related to emotional coping and social preference.

### 3.3. Problematic Internet Use by Language Group and Urbanity

Mean GPIUS-2 total scores did not significantly differ across language groups or between urban and rural adolescents. A small but statistically significant difference was observed in the subscale “Preference for Online Social Interaction”, where Italian-speaking adolescents reported higher scores than their German- and Ladin-speaking peers (*p* = 0.028). No other subscale showed significant variation by language group. Similarly, none of the GPIUS-2 subscales differed significantly by urban versus rural residency, suggesting that problematic Internet use patterns in South Tyrol are relatively consistent across residential and cultural settings.

### 3.4. Correlates of Problematic Internet Use

Bivariate correlations between the GPIUS-2 total score, its subscales, and selected psychological and family related measures are shown in Table 3.

The GPIUS-2 total score showed moderate positive correlations with symptoms of generalized anxiety (SCARED: r = 0.405, *p* < 0.001) and depression (PHQ-2: r = 0.419, *p* < 0.001). Among family related measures, the total MSPSS score and all its subdomains (Family, Friends, Others) were weakly but significantly negatively correlated with the GPIUS-2 total score (r range = −0.096 to −0.112; all *p* < 0.001). In contrast, no significant association was found between GPIUS-2 scores and the Family Affluence Scale (FAS III).

A similar pattern emerged across the five GPIUS-2 subscales (Table 2). All were positively correlated with anxiety and depression symptoms (r range = 0.279–0.431, all *p* < 0.001) and negatively correlated with perceived social support, although the strength of the associations varied. Notably, the “Mood Regulation” subscale had the strongest correlations with psychological distress (SCARED: r = 0.420; PHQ-2: r = 0.431). The “Preference for Online Social Interaction” subscale was most strongly negatively correlated with all MSPSS subdomains (r = −0.128 to −0.164, *p* < 0.001). Only the “Compulsive Internet Use” subscale showed no significant association with any of the family- or support-related variables.

Overall, these results support the relevance of psychological distress as a consistent correlate of PIU and highlight perceived family and social support as potential protective factors, albeit with smaller effect sizes.

### 3.5. Problematic Internet Use and Demographic or Contextual Factors

To explore the potential sociodemographic and contextual correlates of problematic Internet use, the GPIUS-2 total and subscale scores were examined in relation to migration background, family structure, residential environment, parental education, financial stress, and digital family communication (Table 4).

No significant associations were found between the GPIUS-2 total or subscale scores and the binary indicators of migration background, or single parenthood. Similarly, parental education, categorized according to the CASMIN schema, was not significantly associated with any of the PIU domains (Table 5).

In contrast, two contextual variables showed relevant associations. Perceived burden due to price increases, used as a proxy for subjective financial stress, was significantly and positively correlated with the GPIUS-2 total score (ρ = 0.148, *p* < 0.001) and all five subscales (ρ range = 0.117–0.169, all *p* < 0.001). This suggests that adolescents experiencing greater economic strain may be more prone to problematic digital behaviors.

The use of digital parental control tools (Yes/No) was not associated with the total GPIUS-2 score but showed small yet significant associations with several subscales: Preference for Online Social Interaction (*p* = 0.003), Mood Regulation (*p* = 0.004), Cognitive Preoccupation (*p* < 0.001), and Compulsive Internet Use (*p* = 0.010). These findings may indicate selective relationships between parental control measures and specific facets of problematic Internet use.

### 3.6. Multivariable Predictors of Problematic Internet Use

To identify independent predictors of problematic Internet use among adolescents, a multiple linear regression analysis was conducted with the total GPIUS-2 score as the dependent variable (Table 6). Predictor variables were selected based on significant associations observed in bivariate analyses and included age, gender, psychological distress (SCARED score), perceived family support (MSPSS total score), subjective economic burden, and the use of digital parental control tools.

Due to a high correlation between SCARED and PHQ-2 scores, only the SCARED score was retained in the model to avoid multicollinearity. The variable reflecting the use of digital parental control tools was excluded, as this item was not assessed for adolescents aged 18–19 years. The final model explained 17.4% of the variance in the GPIUS-2 total score (R^2^ = 0.174), with acceptable residual independence (Durbin–Watson statistic = 1.903).

Among the included variables, anxiety symptoms (SCARED score) emerged as the strongest predictor (β = 0.349, *p* < 0.001), with higher anxiety associated with higher GPIUS-2 scores. Older age was also independently associated with increased problematic Internet use (β = 0.095, *p* < 0.001). In contrast, perceived family support (MSPSS total score) was negatively associated with the GPIUS-2 score (β = −0.052, *p* = 0.032), indicating a modest protective effect. Subjective economic burden, assessed through concern about price increases, was positively associated with problematic Internet use (β = 0.083, *p* = 0.001). All predictors reached statistical significance and contributed independently to the model after adjusting for confounding.

## 4. Discussion

This study examined the prevalence and correlates of PIU in a large sample of adolescents aged 11–19 years, focusing on psychological distress, family environment, and socioeconomic stressors. Consistent with previous research, PIU levels increased with age, and the subdimension “Mood Regulation” was significantly higher among female adolescents, suggesting that emotional coping plays a particularly relevant role in their online behavior. In line with the CIUT, the findings highlight that depressive and anxiety symptoms were moderately associated with PIU, while indicators of perceived family support were inversely related to PIU. Notably, socioeconomic variables such as parental education and family affluence showed no significant associations, whereas subjective financial strain and the use of digital parental control tools were positively related to higher PIU scores. These findings underscore the role of psychological and perceived contextual burdens in shaping maladaptive digital behaviors in adolescence, more so than structural demographic characteristics do.

### 4.1. Psychological Distress as the Key Driver of Problematic Internet Use

Consistent with the CIUT ([31]), the findings confirm that psychological distress, particularly anxiety and depression symptoms, is a central correlate of PIU among adolescents. In the South Tyrol sample, the strongest associations with the total GPIUS-2 score were observed for depressive symptoms (PHQ-2) and generalized anxiety symptoms (SCARED), with both demonstrating moderate and statistically significant correlations. These findings align with a substantial body of international research showing that psychological distress is not only associated with higher PIU but also predicts its onset and severity ([42]; [2]; [54]; [39]; [16]).

Beyond simple associations, prior longitudinal studies suggest a bidirectional relationship: psychological distress increases the likelihood of PIU, which in turn may worsen mental health over time ([54]; [39]). This underscores the potential for reinforcing the cycles of digital coping and psychological burden. Theoretical frameworks and empirical data support the role of maladaptive coping mechanisms, such as rumination and emotional avoidance, in driving adolescents toward excessive online engagement ([27]; [5]). Furthermore, research highlights moderating and mediating mechanisms; for instance, emotional intelligence appears to buffer this relationship, while self-stigma and rumination may exacerbate it ([4]; [36]).

In this study, the “Mood Regulation” subscale of the GPIUS-2 demonstrated the strongest correlation with psychological distress, particularly among female adolescents. This suggests that emotional regulation through online activities may be a core behavioral pathway linking internalizing symptoms and PIU. Importantly, this domain-specific vulnerability reinforces the conceptualization of PIU as a compensatory behavior aimed at mitigating negative affect ([27]; [24]).

Overall, the results substantiate previous international findings and highlight psychological distress as the most prominent driver of PIU among South Tyrolean adolescents.

### 4.2. Family Support and Digital Control Tools

This study observed a modest but statistically significant inverse association between perceived family support and PIU among adolescents, consistent with earlier research demonstrating that positive family dynamics offer some protection against excessive or maladaptive digital behavior. As shown here, adolescents who reported higher perceived emotional support from family members (as measured by the MSPSS) tended to score slightly lower on the GPIUS-2 total and subscale scores, particularly in domains such as mood regulation and preference for online social interaction. These findings echo a growing body of international evidence highlighting the protective role of supportive parent–child relationships, open family communication, and emotional availability in mitigating the risk of PIU ([12]; [44]; [52]; [40]; [22]). The mechanisms proposed in these studies include reduced adolescent hostility, loneliness, and fear of missing out, as well as improved coping with stressors ([52]; [28]). However, consistent with prior reviews ([41]), the effect sizes in our data were small, suggesting that while family support is beneficial, it may not be sufficient to prevent PIU in the presence of other psychosocial vulnerabilities.

In contrast, an unexpected positive association emerged between the use of parental digital control tools, such as Apple Family Sharing and Google Family Link, and higher GPIUS-2 scores. This finding suggests that adolescents in households with stricter digital supervision may already exhibit problematic Internet behaviors, prompting parental control as a reactive measure. This pattern is consistent with reverse causality. Prior studies also indicate that restrictive parental mediation, including enforced time limits or surveillance tools, may inadvertently undermine adolescent autonomy and trust, potentially leading to resistance, covert online behavior, or increased stress, all of which may exacerbate PIU ([41]; [16]). Moreover, as noted in recent meta-analytic and profile-based research, the effectiveness of parental regulation strongly depends on style and context: active mediation and collaborative discussions about digital content tend to show more promising outcomes than rigid enforcement of screen-time rules ([41]; [48]).

Taken together, these results underscore the complexity of family influences on adolescent PIU. Although perceived emotional support remains a modest but consistent protective factor, digital restriction strategies may not achieve their intended effect. These findings suggest that prevention efforts should emphasize strengthening family relationships and fostering communication-based digital education rather than relying solely on technical control tools.

### 4.3. Subjective Economic Burden and Structural Socioeconomic Status

The current study found that adolescents’ perceived economic burden, operationalized as self-reported stress due to price increases, was significantly associated with higher GPIUS-2 scores across all subdomains of problematic Internet use. In contrast, traditional structural socioeconomic status indicators, such as parental education (CASMIN classification) and family affluence (FAS III), showed no significant associations with PIU.

This aligns with an increasing empirical evidence suggesting that subjective perceptions of financial strain are more sensitive predictors of adolescent psychological outcomes and risk behaviors than objective socioeconomic status measures ([34]; [49]). Adolescents’ experience of financial insecurity may heighten emotional distress and reduce access to adaptive coping resources, fostering compensatory Internet behavior. In our findings, this subjective burden, despite being assessed with a single item, showed consistent correlations with both the total PIU score and all the GPIUS-2 subscales.

In contrast, parental education and material wealth, as assessed via the CASMIN and FAS III, respectively, were not significantly related to PIU in this sample. Prior studies have similarly reported weak or inconsistent associations between structural socioeconomic status indicators and adolescent Internet use patterns ([43]; [45]). These discrepancies suggest that structural socioeconomic status alone may be insufficient to explain digital vulnerability, especially in relatively affluent or socially cohesive regions, such as South Tyrol.

Moreover, no significant differences were observed in PIU scores based on migration background, family structure (e.g., single-vs. two-parent households), or residential urbanity. This may reflect the degree of social integration and equal access to digital environments across demographic strata in the South Tyrolean context.

Although our analyses found no significant differences in PIU scores by language group or urban-rural residence, these variables remain critical for understanding regional behavioral patterns in PIU. Research shows that cultural, regional, and linguistic differences can influence adolescents’ digital behaviors and PIU vulnerability. Cultural identity, norms, parental monitoring, and technology access shape PIU patterns across languages and ethnic groups ([6]). Urban adolescents may face higher PIU risks because of increased online exposure ([3]). The absence of differences in our South Tyrolean data may reflect a cohesive sociocultural environment, highlighting the need for localized research to detect subtle interactions.

Cross-national evidence is consistent with this pattern: studies from Spain report that depressive and anxiety symptoms are the strongest correlates of adolescent PIU, while SES indicators show weak or inconsistent associations ([45]; [47]). Large samples from China similarly find psychological distress as a primary predictor, with family and structural factors exerting smaller effects ([37]; [17]; [56]). Evidence from Latvia integrates behavioral and parenting factors but again underscores the role of internal distress and maladaptive cognitions ([51]). Meta-analytic and profile-based work further indicates that family functioning tends to buffer rather than replace the influence of psychological distress on PIU ([26]), and that restrictive parental control can be counterproductive depending on style and context ([41]; [16]). Overall, international findings align with our results by placing psychological distress ahead of structural socioeconomic markers, with family processes exerting modest, context-dependent effects.

### 4.4. Strengths and Limitations

This study offers key strengths that contribute to the evidence of adolescent PIU. First, the large sample of over 1500 adolescents, recruited province-wide across a diverse multilingual region enables meaningful subgroup analyses and enhances generalizability within South Tyrol. Second, this study used validated international instruments to assess PIU, mental health, and family context, allowing reliable comparisons with prior research. Third, the integration of structural and subjective socioeconomic indicators, along with psychological and family level variables, provides a refined understanding of the potential risk and protective factors.

This study has several limitations must be acknowledged. The cross-sectional design prevents inferring causality and limits the interpretation of temporal relationships between PIU and its correlates. All data were self-reported, which may have introduced reporting bias for sensitive mental health indicators or digital behaviors. PIU assessment was based on questionnaire data rather than behavioral tracking or clinical diagnosis, potentially not capturing the full range of problematic uses. A notable limitation is that although 2554 adolescents provided consented self-reports, only 1550 participants completed the full set of GPIUS-2 items. This drop-off may introduce non-response bias if systematic differences exist between respondents with and without complete data on Internet use. Moreover, no statistical weighting was applied beyond age and sex distributions, which may limit the generalizability across other sociodemographic subgroups.

The lack of differences across language groups and urbanity may reflect the relatively cohesive and well-integrated regional environment of South Tyrol, underscoring the need for further cross-regional comparative studies. The interpretation of language group comparisons requires caution due to the pronounced imbalance in subgroup sizes. The smaller Ladin and “Other” groups substantially limit statistical power and increase the likelihood of Type II error, potentially masking subtle differences. These unequal group sizes reflect the true demographic composition of South Tyrol’s multilingual population rather than sampling bias; therefore, similar proportions are expected in future COP-S survey waves. Although one-way ANOVA is robust to moderate violations of variance homogeneity, results for smaller groups should be interpreted with care.

### 4.5. Implications and Future Research

The findings emphasize that prevention and intervention strategies for adolescent problematic Internet use should prioritize mental-health promotion and emotional coping skills. Integrating brief mental-health screening and early psychosocial support into school- and community-based prevention programs could help identify adolescents at risk before maladaptive use patterns become entrenched.

At the policy level, these results suggest the value of embedding digital-behavior prevention within adolescent mental-health frameworks that coordinate education, public health, and family services. Such integration would facilitate early detection and provide a pathway from screening to timely psychosocial intervention.

From a research perspective, future studies should extend beyond confirming causal pathways to examine interaction effects among psychological distress, family communication, cultural identity, and socioeconomic stressors in shaping digital behaviors. Mixed-methods and cross-cultural comparative designs are particularly needed to capture contextual moderators and mediators across diverse societies and to understand how cultural norms influence online coping and regulation.

Finally, in the educational domain, digital-media literacy should move beyond rule enforcement and technical restrictions toward cultivating emotional literacy, self-regulation, and participatory approaches that actively engage families and reflect cultural diversity. Such emotionally informed, culturally sensitive education may prove more sustainable than purely control-based strategies and can strengthen adolescents’ resilience in navigating digital environments.

## 5. Conclusions

Adolescence is a period of significant emotional and social transformation, increasingly shaped by digital environments that present opportunities for connection and risks. Understanding why some adolescents develop problematic Internet use, while others do not, is essential for safeguarding mental health in a hyperconnected generation. The present findings contribute to this endeavor by showing how psychological vulnerability, family relationships, and subjective stress interact within a multilingual European context. They emphasize that interventions must extend beyond screen-time regulation to address emotional needs and coping mechanisms that drive youth online. This study provides robust, regionally grounded evidence that psychological distress is the most prominent driver of PIU among adolescents, outweighing structural, demographic, or socioeconomic predictors. Although perceived family support offers modest protection, the use of digital parental control tools may be reactive and, in some cases, counterproductive. Notably, adolescents’ subjective experience of financial burden, rather than objective socioeconomic indicators, was associated with higher PIU scores, highlighting the relevance of perceived stress in digital behavior. These findings support the application of CIUT and emphasize the need for mental health-informed, family-sensitive, and emotionally attuned approaches to digital health promotion in adolescence.

## Figures and Tables

**Figure 1 behavsci-15-01534-f001:**
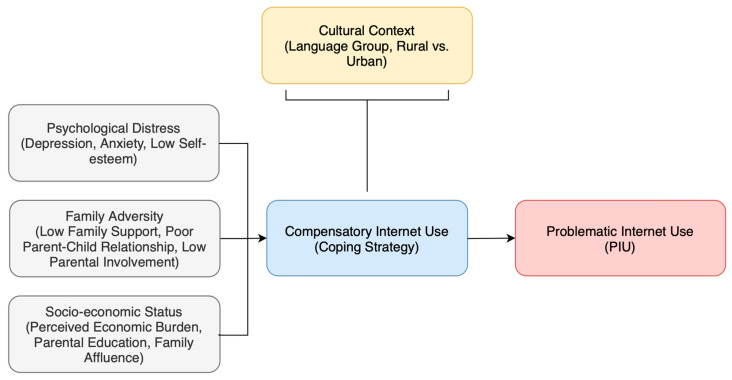
Conceptual framework based on Compensatory Internet Use Theory (CIUT) applied to South Tyrolean adolescents. This conceptual model illustrates the pathways to problematic Internet use (PIU) among adolescents in South Tyrol based on CIUT.

**Figure 2 behavsci-15-01534-f002:**
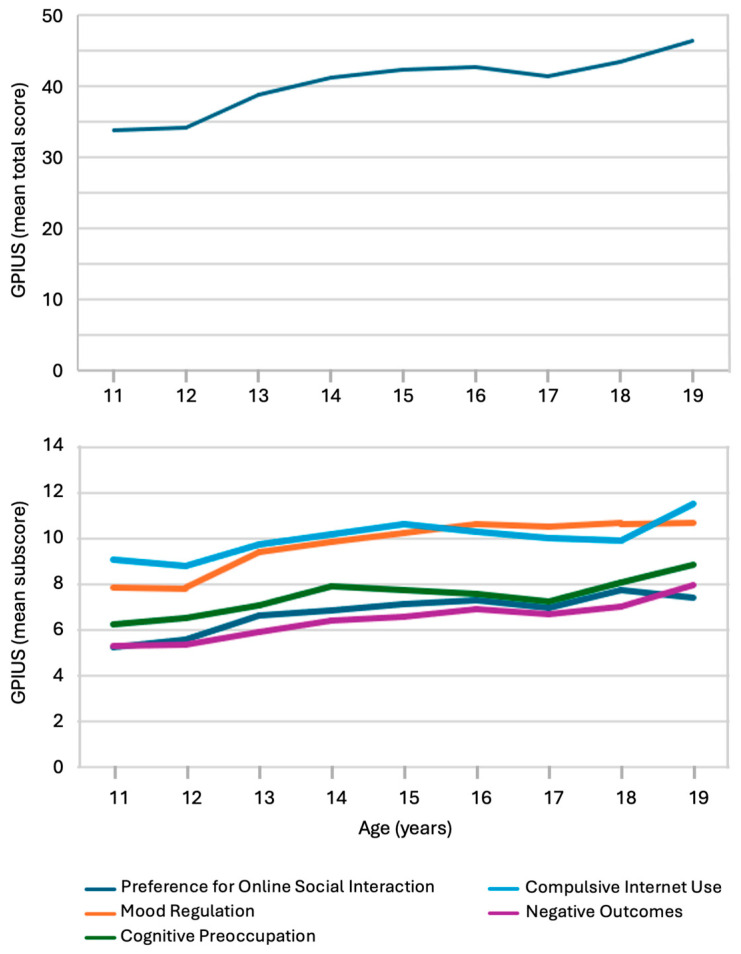
Age-related increase in total and subscale scores of the Generalized Problematic Internet Use Scale 2 (GPIUS-2) in adolescents aged 11–19 years. (**Top panel**): Mean total GPIUS-2 scores by age group. (**Bottom panel**): Mean scores of the five GPIUS-2-2 subdimensions by age. Each subdimension reflects a specific aspect of problematic Internet use. Values are based on unweighted self-reported data from adolescents (*n* = 1550) participating in the COP-S 2025 survey in South Tyrol.

**Table 1 behavsci-15-01534-t001:** Sample characteristics of adolescents aged 11–19 years included in the COP-S 2025 survey (*n* = 1550).

Variable	Category	*n*	%
Age group	11–14 years	809	52.2%
	15–19 years	740	47.8%
Gender ^1^	Female	772	49.8%
	Male	777	50.1%
Family language	German	1264	81.8%
	Italian	222	14.4%
	Ladin	41	2.7%
	Other	18	1.2%
Migration background	Yes	129	8.5%
	No	1392	91.5%
Parental education (CASMIN)	Low (primary/lower secondary)	284	18.4%
	Medium (upper secondary)	646	41.7%
	High (tertiary)	610	39.6%
Family structure	Two-parent household	1359	88.1%
	Single-parent household	183	11.9%
Urbanity	Urban	447	28.8%
	Rural	1103	71.2%
Subjective economic burden	Not at all burdened (1) ^2^	25	1.6%
	Slightly to moderately burdened (2–3)	602	38.9%
	Strongly to extremely burdened (4–5)	919	59.5%
Family Affluence Scale (FAS III)	Low	268	17.5%
	Medium	870	56.8%
	High	395	25.8%
Use of digital parental controls ^3^	Yes	734	53.1%
	No	628	45.4%
Use of parental help with school	Never/rarely	263	17.0%
	Sometimes/often/always	1155	74.5%
	Never asked for help	130	8.4%

^1^ One respondent identified as diverse; this case was not included in gender-stratified analyses. ^2^ Likert scale in parenthesis. ^3^ Only 11–17 years (*n* = 1383).

**Table 2 behavsci-15-01534-t002:** Mean scores and standard deviations of the Generalized Problematic Internet Use Scale 2 (GPIUS-2) total and subscale scores by sex and age group among South Tyrolean adolescents (*n* = 1550).

GPIUS-2 Score	Score, Mean (SD)	Gender			Age		
		Females (Mean)	Males (Mean)	*p*-Value	11–14 Years	15–19 Years	*p*-Value
Total score	39.61 (19.49)	39.91	39.30	n.s.	36.89	42.59	<0.001
Preference for Online Social Interaction	6.61 (4.50)	6.67	6.55	n.s.	6.03	7.24	<0.001
Mood Regulation	9.58 (5.60)	10.18	8.97	<0.001	8.73	10.51	<0.001
Cognitive Preoccupation	7.31 (4.60)	7.24	7.37	n.s.	6.94	7.70	<0.001
Compulsive Internet Use	9.86 (5.20)	9.76	9.97	n.s.	9.46	10.31	0.001
Negative Outcomes	6.29 (3.90)	6.19	6.39	n.s.	5.78	6.85	<0.001

n.s., not significant.

**Table 3 behavsci-15-01534-t003:** Mean (SD) Generalized Problematic Internet Use Scale 2 (GPIUS-2) total and subscale scores by family language group and urbanity in adolescents (*n* = 1550).

GPIUS-2 Score, Mean (SD)	German (*n* = 1264)	Italian (*n* = 222)	Ladin (*n* = 41)	*p*-Value	Urban (*n* = 447)	Rural (*n* = 1103)	*p*-Value
Total score	39.49 (19.55)	40.50 (19.76)	40.02 (17.84)	n.s.	39.56 (20.39)	39.63 (19.12)	n.s.
Preference for Online Social Interaction	6.49 (4.44)	7.30 (4.87)	7.05 (4.26)	0.028	6.76 (4.71)	6.56 (4.43)	n.s.
Mood Regulation	9.50 (5.67)	10.03 (5.39)	9.88 (5.50)	n.s.	9.62 (5.75)	9.55 (5.55)	n.s.
Cognitive Preoccupation	7.33 (4.64)	7.19 (4.60)	7.34 (4.09)	n.s.	7.25 (4.60)	7.31 (4.61)	n.s.
Compulsive Internet Use	9.91 (5.24)	9.70 (5.10)	9.80 (5.08)	n.s.	9.68 (5.18)	9.93 (5.20)	n.s.
Negative Outcomes	6.27 (3.91)	6.27 (3.81)	5.95 (3.35)	n.s.	6.25 (3.93)	6.27 (3.87)	n.s.

Values are presented as mean and standard deviation (SD); *p*-values are from Kruskal–Wallis tests (language group) and Mann–Whitney U tests (urban vs. rural); n.s., not significant.

**Table 4 behavsci-15-01534-t004:** Spearman correlations between the Generalized Problematic Internet Use Scale 2 (GPIUS-2) total and subscale scores and mental health, perceived social support, and family affluence (adolescents aged 11–19 years, *n* = 1550).

GPIUS-2 Score	SCARED	PHQ-2	MSPSS				FAS III
			Total	Family	Friends	Other	
Total score	0.405 ***	0.419 ***	−0.097 ***	−0.112 ***	−0.112 ***	−0.096 ***	n.s.
Preference for Online Social Interaction	0.291 ***	0.356 ***	−0.151 ***	−0.128 ***	−0.164 ***	−0.151 ***	n.s.
Mood Regulation	0.420 ***	0.431 ***	−0.081 ***	−0.096 ***	−0.074 **	−0.055 *	n.s.
Cognitive Preoccupation	0.279 ***	0.281 ***	−0.084 ***	−0.086 **	−0.084 **	−0.092 ***	n.s.
Compulsive Internet Use	0.316 ***	0.316 ***	n.s.	n.s.	n.s.	n.s.	n.s.
Negative Outcomes	0.354 ***	0.356 ***	−0.101 ***	−0.125 ***	−0.129 ***	−0.105 ***	n.s.

Spearman correlation coefficient. GPIUS-2, Generalized Problematic Internet Use Scale 2; SCARED, Screen for Child Anxiety Related Emotional Disorders, GAD subscale; PHQ-2, Patient Health Questionnaire-2 (depressive symptoms); MSPSS, Multidimensional Scale of Perceived Social Support; FAS III, Family Affluence Scale III. * *p* < 0.05, ** *p* < 0.01, *** *p* < 0.001; n.s., not significant.

**Table 5 behavsci-15-01534-t005:** Associations between Generalized Problematic Internet Use Scale 2 (GPIUS-2) total and subscale scores and selected demographic and contextual factors among adolescents aged 11–19 years (*n* = 1550).

GPIUS-2 Score	Migration Background (Yes vs. No)	Single Parenthood (Yes vs. No)	Parental Education (Low/Med/High)	Worry About Higher Prices (Spearman’s ρ)	Parental Control Tool (Yes vs. No)
Total Score	n.s.	n.s.	n.s.	0.148 ***	n.s.
Preference for Online Social Interaction	n.s.	n.s.	n.s.	0.149 ***	0.003 **
Mood Regulation	n.s.	n.s.	n.s.	0.169 ***	0.004 **
Cognitive Preoccupation	n.s.	n.s.	n.s.	0.122 ***	<0.001 ***
Compulsive Internet Use	n.s.	n.s.	n.s.	0.117 ***	0.010 *
Negative Outcomes	n.s.	n.s.	n.s.	0.139 ***	n.s.

Values represent either non-significant findings (n.s.) or Spearman’s rank correlation coefficients (ρ) for the continuous variables. Migration background, single parenthood, urban residency, and parental education were tested using the Mann–Whitney U or Kruskal–Wallis tests. Worry about higher prices was assessed using a 5-point Likert scale (1 = not burdened to 5 = extremely burdened). The use of digital parental control tools was analyzed using binary response options (yes/no). * *p* < 0.05,** *p* < 0.01, *** *p* < 0.001.

**Table 6 behavsci-15-01534-t006:** Multiple linear regression predicting total GPIUS-2 score from psychological distress, perceived family support, age, and subjective financial burden (*n* = 1550).

Predictor Variable	B (Unstandardized Coefficient)	Standard Error	Β (Standardized Coefficient)	t	*p*-Value	95% CI (Lower–Upper)
(Constant)	20.446	3.660	—	5.587	<0.001	[13.267, 27.625]
Age (in years)	0.790	0.205	0.095	3.859	<0.001	[0.388, 1.191]
Perceived Family Support (MSPSS total score)	−0.715	0.333	−0.052	−2.147	0.032	[−1.368, −0.062]
Anxiety Symptoms (SCARED score)	1.432	0.104	0.349	13.827	<0.001	[1.229, 1.635]
Subjective Financial Burden (Price Increase Concern)	1.307	0.394	0.083	3.316	0.001	[0.534, 2.081]

Abbreviation: CI, confidence interval.

## Data Availability

The data presented in this study are available from the corresponding author upon reasonable requests.

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
