# Peer review of "Problematic Internet Use in Adolescents Is Driven by Internal Distress Rather Than Family or Socioeconomic Contexts: Evidence from South Tyrol, Italy"

_behavsci, 2025, doi:10.3390/bs15111534_

Round 1
Reviewer 1 Report
Comments and Suggestions for Authors
The article examines problematic Internet use among 1,550 adolescents in South Tyrol, Northern Italy, situating it within the broader context of mental health, psychological distress, and emotional regulation. Drawing on a cross-sectional survey of 1,550 adolescents, the study employs robust, validated instruments, including the Generalized Problematic Internet Use Scale 2 (GPIUS-2), the PHQ-2 for depressive symptoms, and the SCARED-GAD for anxiety assessment.
Despite the study’s strong methodological foundation and its valuable contribution to understanding adolescent problematic Internet use, the manuscript would benefit from a major revision before publication, as there are significant issues to be addressed.
1) Lines 60–62: The authors state that “The ‘Corona and Psyche South Tyrol’ (COP-S) 2025 survey offers a perspective for exploring how adolescents in South Tyrol's cultural context may use digital environments in response to perceived burdens.” However, this claim lacks sufficient explanation. The manuscript should clarify how the COP-S survey design, measures, or contextual components specifically enable such exploration. For instance, the authors could elaborate on which variables capture adolescents’ use of digital environments, the nature of “perceived burdens” assessed, and how the survey’s cultural framing supports interpretation within South Tyrol’s multilingual and socio-cultural context. Without this clarification, the statement appears speculative rather than empirically grounded.
2) The manuscript presents clear research questions but omits explicit hypotheses, which is a notable omission for a quantitative study grounded in established prior research. Given the existing literature linking emotional distress to problematic Internet use, the authors should formulate and state testable hypotheses that can be empirically evaluated. For example, a primary hypothesis might predict that higher levels of depression and anxiety are associated with increased GPIUS-2 scores, while a secondary hypothesis could anticipate minimal associations between demographic variables and PIU. Including such hypotheses would strengthen the study’s methodological rigor, clarify the analytical framework, and align the work with conventional standards for quantitative research design.
3) Lines 112–113: The authors claim that “The sample approximates the regional distribution of school-aged youth by age and gender based on official provincial statistics.” However, the manuscript does not provide details on how this approximation was calculated. The authors need to explain the methods used to compare their sample demographics with the official provincial statistics—whether through quota sampling, weighting procedures, or statistical tests assessing representativeness.
4) Lines 118–119 include statements that are repeated from the next paragraph, which appears to be a mistake. The authors should remove the duplicated sentences—“Adolescents self-reported their age and sex. The language group was based on parental reports of the home language spoken with the child”
5) Lines 144-145. It is advisable to mention the specific items of the PHQ-2 directly in the text, given its concise 2-item structure. Including the actual items would clarify exactly what aspects of depression were assessed and enhance transparency.
6) Lines 154-156 - Here lies the major issue of this work: Self-esteem was assessed using a single item on body image perception, asking adolescents to rate their body from “far too thin” to “far too fat.” This approach raises a significant concern. Evaluating self-esteem through a body size perception scale is problematic because it implicitly assigns value judgments—suggesting that being “too thin” or “too fat” carries a positive or negative connotation—without directly measuring self-worth or self-acceptance. A more appropriate measure would capture perceptions in terms of comfort or positivity toward one’s body rather than size-based labels. Given this conceptual flaw and the lack of significant associations found with this variable, it is strongly recommended that this measure be removed from the study.
7) The authors should include at least one example item for each of the five subdimensions of problematic Internet use (PIU), since it will help readers better understand how each dimension was measured.
8) Given the large disparity in sample sizes among language groups—German (1264, 81.8%), Italian (222, 14.4%), Ladin (41, 2.7%), and Other (18, 1.2%)—there are legitimate concerns about the robustness of statistical comparisons across these groups. The small sample sizes for Ladin and Other speakers limit statistical power and increase the risk of Type II error, potentially masking true differences or inflating significance in smaller groups (e.g., the Italian group’s higher score on “Preference for Online Social Interaction”). Additionally, such uneven group sizes may violate assumptions of homogeneity of variance in parametric tests. Therefore, conclusions about the lack of significant differences across language groups and about the general consistency of problematic Internet use should be interpreted cautiously, with acknowledgement of these methodological constraints. It may be advisable for the authors to discuss these limitations explicitly and consider supplementary analyses or more balanced sampling in future studies.
9) The manuscript should clarify the analytical approach used to examine the relationships between GPIUS-2 total and subscale scores and the sociodemographic/contextual variables such as migration background, family structure, residential environment, parental education, financial stress, and digital family communication. Specifically, the authors need to detail whether these associations were explored using bivariate analyses, multivariable regression models, or other statistical techniques.
10)The analysis of the relationship between parental use of digital control tools and problematic Internet use should be conducted using mean comparison techniques rather than simple association measures. Specifically, comparing the mean GPIUS-2 scores between groups defined by the presence or absence (or levels) of parental control tools using t-tests would be more appropriate.
Reviewer 2 Report
Comments and Suggestions for Authors
See file

Author Response
We sincerely thank the reviewer for this very positive and encouraging evaluation. We are pleased that the study design, methodological transparency, and clarity of presentation were well received. We greatly appreciate the reviewer’s supportive feedback and acknowledgment of the paper’s contribution to the literature on adolescent problematic Internet use beyond U.S. contexts.
Reviewer 3 Report
Comments and Suggestions for Authors
Dear authors,
here are some recommendations for you:
- "Adolescents self-reported their age and sex. The language group was based on 118
parental reports of the home language spoken with the child." - You repeat this twice. - In Table 1 you dropped out malesYou missed data about Cronbach's alpha of instruments you used.
- My personal belief is that the main mistake you made in the settings of your paper is the use of so many "instruments" with a small number of items, at least three of them.
- In the Discussion, it would be wise to add more researches from other countries, since you didn't get this regional difference.
- I disagree with Implications of future research. Please reconsider the improvement of it.
- Also, I believe that the weakest point of the manuscript is conclusion. It should be more inspirative and longer, according to your results.
Round 2
Reviewer 1 Report
Comments and Suggestions for Authors
The authors have thoroughly and carefully addressed all the comments raised in the first review, making the necessary revisions and additions accordingly. Therefore, I find no reason to withhold my recommendation for the acceptance of this article for publication.
Author Response
We sincerely thank the editor for this positive evaluation and for recognizing our revisions. We appreciate the opportunity to improve the manuscript and are grateful for the recommendation for acceptance.
Reviewer 3 Report
Comments and Suggestions for Authors
Dear Authors,
1. I am sorry, but this paragarph in second version of your manuscript is missing, so as the literature used in it, in the references part.
"Cross-national evidence is consistent with this
pattern: studies from Spain report that
depressive and anxiety symptoms are the
strongest correlates of adolescent PIU, while
SES indicators show weak or inconsistent
associations (Nogueira-López et al. 2023;
Piqueras et al. 2024). Large samples from
China similarly find psychological distress as a
primary predictor, with family and structural
factors exerting smaller effects (Li et al. 2019;
Chi et al. 2020; Zhang et al. 2022). Evidence
from Latvia integrates behavioral and parenting
factors but again underscores the role of internal
distress and maladaptive cognitions (Sebre et
al. 2020). Meta-analytic and profile-based work
further indicates that family functioning tends to
buffer rather than replace the influence of
psychological distress on PIU (Fumero et al.
2018), and that restrictive parental control can
be counterproductive depending on style and
context (Lukavská et al. 2022; Chen and Fan
2024). Overall, international findings align with
our results by placing psychological distress
ahead of structural socioeconomic markers, with
family processes exerting modest, contextdependent effects."
2. In the Conclusion, this is missing too:
"To improve the conclusions, the following text
was added as a first paragraph:
Adolescence is a period of significant emotional
and social transformation, increasingly shaped
by digital environments that present
opportunities for connection and risks.
Understanding why some adolescents develop
problematic Internet use, while others do not, is
essential for safeguarding mental health in a
hyperconnected generation. The present
findings contribute to this endeavor by showing
how psychological vulnerability, family
relationships, and subjective stress interact
within a multilingual European context. They
emphasize that interventions must extend
beyond screen-time regulation to address
emotional needs and coping mechanisms that
drive youth online."
2. Please, check this one also:
"Concrete policy implications: Mental health
screening is now framed within school- and
community-based prevention programs,
emphasizing early psychosocial support for
at-risk adolescents.
• Enhanced research focus: The section now
specifies that future studies should examine
interaction effects among psychological
distress, family communication, cultural
identity, and socioeconomic stressors.
• Cross-cultural perspective: A new call for
mixed-methods and comparative
international research put the paper in
context with broader literature and
Reviewer’s suggestion to include more
studies from other countries.
• Educational dimension: The concept of
digital media education was expanded
beyond “rule enforcement” to include
emotional literacy, self-regulation, and
participatory approaches that involve
families and cultural adaptation."
Author Response
We sincerely apologize for the inadvertent omission of the updated text in Section 4.5 (“Implications and Future Research”) in the previous submission. The section has now been fully revised and reinserted as intended, incorporating all elements outlined in our prior response — including the policy implications, enhanced research focus, cross-cultural perspective, and educational dimension.
We thank the reviewer for bringing this to our attention and for their careful rereading of the manuscript. With these corrections, all requested revisions have now been completed in the present version (Revision 2).
Round 3
Reviewer 3 Report
Comments and Suggestions for Authors
Thank you, everything is fine now.